# Estrogens Play a Critical Role in Stress-Related Gastrointestinal Dysfunction in a Spontaneous Model of Disorders of Gut–Brain Interaction

**DOI:** 10.3390/cells11071214

**Published:** 2022-04-04

**Authors:** Alison Accarie, Joran Toth, Lucas Wauters, Ricard Farré, Jan Tack, Tim Vanuytsel

**Affiliations:** 1Translational Research Center for Gastrointestinal Disorders (TARGID), Department of Chronic Diseases and Metabolism (ChroMetA), KU Leuven, 3000 Leuven, Belgium; alison.accarie@gmail.com (A.A.); joran.toth@kuleuven.be (J.T.); lucas.wauters@kuleuven.be (L.W.); ricard.farre@kuleuven.be (R.F.); jan.tack@kuleuven.be (J.T.); 2Department of Gastroenterology and Hepatology, University Hospitals Leuven, 3000 Leuven, Belgium

**Keywords:** multifactorial DGBI model, β-estradiol, disorders of gut–brain interaction, irritable bowel syndrome, mast cells, visceral hypersensitivity, sex differences

## Abstract

Background: Disorders of the gut–brain interaction (DGBI), such as irritable bowel syndrome and functional dyspepsia, are more prevalent in women than in men, with a ratio of 2:1. Furthermore, stressful life events have been reported as one of the triggers for symptoms in DGBI patients. Methods: Here, we studied the effect of an early-life stressor (maternal separation (MS)) on jejunal and colonic alterations, including colonic sensitivity and immune cells infiltration and activation in a validated spontaneous model of DGBI (BBDP-N), and investigated the involvement of β-estradiol on stress-worsened intestinal alterations. Results: We found that maternal separation exacerbated colonic sensitivity and mast cell and eosinophil infiltration and activation in females only. Ovariectomy partially rescued the stress phenotype by decreasing colonic sensitivity, which was restored by β-estradiol injections and did not impact immune cells infiltration and activation. Stressed males exposed to β-estradiol demonstrated similar intestinal alterations as MS females. Conclusion: Estrogen plays a direct critical role in colonic hypersensitivity in a spontaneous animal model of DGBI, while for immune activation, estrogen seems to be involved in the first step of their recruitment and activation. Our data point towards a complex interaction between stress and β-estradiol in DGBI.

## 1. Introduction

The impact of psychological stress on gastrointestinal (GI) physiology has gained growing interest over the last decade. A multitude of studies have demonstrated that, in disorders of gut–brain interaction (DGBI)—such as irritable bowel syndrome (IBS) and functional dyspepsia (FD), previously referred to as functional gastrointestinal disorders—psychological stress and anxiety are common, influencing the onset of symptoms and predicting clinical outcome [1,2]. Several preclinical studies have investigated the role of stress—acute or chronic, applied in adulthood or during early life—leading to the conclusion that healthy animals submitted to stress developed intestinal alterations, such as increased intestinal permeability, visceral hypersensitivity, mast cell infiltration and increased protease activity, which were more pronounced in females than males [3,4,5]. In women, symptoms usually appear around the time of puberty and increase in severity in the early adult years, suggesting an involvement of the reproductive hormones in their pathophysiology. Moreover, 40% of women with IBS describe an exacerbation of symptoms around the menses, and one third of healthy women experience GI symptoms during this period. In an interesting study, Whitehead and collaborators found that women with DGBI were affected to a greater degree by menstruation in terms of GI symptoms than women without DGBI, suggesting a different reaction to ovarian hormonal fluctuations in case of DGBI [6,7]. Several preclinical studies have demonstrated an important role for estrogen in the pathophysiology of stress-induced GI alterations, especially through spinal regulation of serotonin, enhanced expression of NMDA and estrogen receptors [8,9,10,11]. Estrogens execute their effect through their receptors—ERα, Erβ and G protein-coupled estrogen receptor (GPER). Depending on the model and inflammatory state, agonists for those receptors induce a pro- or anti-nociceptive effect. In the GI tract, they have been linked to visceral pain [4,12], motility [13] and inflammation [14].

However, most preclinical studies on DGBI have usually used one exogenous factor as a trigger for DGBI-like intestinal alterations, while an increasing amount of evidence has shown that the etiology of DGBI is multifactorial and most likely involves a combination of several factors, such as psychosocial history, food, genetic predisposition and microbiota. We have characterized the normoglycemic BioBreeding (BBDP-N) rat model as a spontaneous model for DGBI, presenting features of gastrointestinal sensorimotor dysfunction which are reminiscent of both functional dyspepsia (FD) and IBS [15,16,17]. This model presents the advantage to be spontaneous with intestinal alterations appearing during (early) adulthood, affecting both the jejunum and colon [15,16], which allows us to study the effect of factors such as early-life stress on the background of a DGBI model. However, most studies on the gastrointestinal phenotype of this model have been performed in males, which does not reflect the sex distribution in human DGBI cases. Based on the results of our previous studies, we worked with rats at 90 days old, an age at which jejunal alterations are already present (e.g., increased permeability, immune cell infiltration) and colonic alterations are starting to appear (mast cell and eosinophil infiltration) [17].

Here, we studied the impact of an early-life stressor in both sexes of this spontaneous rat model with a focus on peripheral, gastrointestinal alterations, including eosinophil and mast cell infiltration and activation and colonic sensitivity. In a second part, we further investigated the specific role of β-estradiol in these stress-related intestinal alterations.

## 2. Materials and Methods

### 2.1. Animals

Breeding pairs of normoglycemic diabetes-prone (BBDP-N) rats were obtained from the Ottawa Hospital Research Institute and further bred in the conventional animal facility of the KU Leuven, Belgium. Rats were housed on a 12 h light/dark cycle with ad libitum access to drinking water and standard chow. Glycemia was measured on tail blood weekly using a OneTouch^®^Verio^®^ glucometer (LifeScan, Diegem, Belgium) and only rats remaining normoglycemic throughout the study were included in the results analysis.

### 2.2. Experimental Design

#### 2.2.1. Protocol 1: Effect of Maternal Separation on GI Phenotype of the Adult BBDP-N Rat

Newborn male and female BBDP-N rats were included in a study using a maternal separation (MS) stress model. BBDP-N litters were divided into a stressed and a nonhandled (NH) group. For the stress group, pups were separated from their dam for 3 h/day from postnatal day (PND) 2 till PND14. In the nonhandled group, pups were left undisturbed until weaning. At 21 days, all pups were weaned normally. At 80 days, rats were implanted with a telemetric transmitter, and 5–7 days later, colonic sensitivity was evaluated. Jejunal and colonic tissue were harvested and processed for histology or snap-frozen in liquid nitrogen for further experiments (Appendix A). The snap frozen pieces were separated in two parts, one contained the mucosa and the submucosa and the second one the deeper layers (muscularis propria, myenteric plexus and serosa). These samples were obtained by peeling off the muscle layer from the superficial layers.

#### 2.2.2. Protocol 2: Estrogen Involvement in Maternal-Separation-Induced GI Alterations in Adult BBDP-N Rats

A similar MS paradigm as in protocol 1 was used. At 80 days, colonic sensitivity was tested (baseline). After sensitivity testing, the females were randomized to either ovariectomy or sham surgery. Colonic sensitivity was evaluated 15 days after surgery, after which the rats were euthanized, and tissue was harvested (Appendix A). The male rats were randomized to receive 4 subcutaneous injections (one every 4 days) of either β-estradiol or vehicle (safflower oil). β-estradiol 3-benzoate was dissolved at a concentration of 0.5 µg/µL and, at each injection, the rats received 50 µg dissolved in 100 µL of vehicle [18] (Appendix A). In a second part, another batch of MS ovariectomized females were injected with either β-estradiol or vehicle 15 days after ovariectomy and the baseline recording for the visceromotor response as described for the males. For each group, colonic sensitivity was evaluated again 24 h after the second (females) or third (males) injection. After the third (females) or fourth (males) injection, animals were euthanized and jejunal and colon tissue were harvested (Appendix A).

### 2.3. Colonic Sensitivity

Colonic sensitivity was assessed by recording abdominal muscle activity in response to isobaric colorectal distensions. A Physiotel ETA-F10 telemetric transmitter (Data Sciences International, MC, ‘s-Hertogenbosch, The Netherlands) was inserted subcutaneously and sutured in the abdominal muscles. On the day of the measurement, rats were lightly anesthetized with Isoflurane to allow smooth insertion of the distension probe into the colon with the distal end of the balloon at 1.5 cm from the anal margin. Graded isobaric colorectal distensions increasing from 15 to 60 mmHg were applied using a Barostat Distender series IIR (G&J Electronics, Toronto, ON, Canada) with 15 mmHg increment steps, with a duration of 20 s for each step and 4 min intervals between distensions. The visceromotor response (VMR) was measured and quantified using DataQuest software (Data Sciences International, MC, ‘s-Hertogenbosch, The Netherlands). For analysis, the mean value of the resting electromyography (EMG) signal 20 s before distension (i.e., basal activity) was subtracted from the mean value of the EMG signal evoked during the 20 s distension. To account for the different balloon and colonic diameters, the data were expressed as the linear regression of the VMR in response to wall tension stimulation, calculated as previously described [16].

### 2.4. Surgery

Two days after the first colonic measurement, females were randomly assigned to the ovariectomy or sham group. Rats were anesthetized with isoflurane and a dorsal approach was used to remove the two ovaries. The skin and the abdominal cavity were opened with a 1 cm incision, the ovary was lifted and after ligature of the uterine corn just below the ovary, the latter was cut, and the ovaries removed, after which the abdominal cavity and skin were closed. For the sham-operated rats, the same procedure was followed except that the ovaries were lifted from the abdominal cavity for a few seconds and placed back afterwards.

### 2.5. Histology

Sections of the distal colon and jejunum were fixed overnight in paraformaldehyde 3.7 % (Sigma-Aldrich, Diegem, Belgium) and embedded in paraffin. Sections of 5 µm were cut with a microtome. Specific quantification of eosinophils was performed using Chromotrope 2R staining [16,19]. Briefly, following deparaffinization with xylene and rehydration with serial ethanol dilutions, slides were immersed for 1.5 h in a solution containing Phenol (106.2 mM; Sigma-Aldrich, Diegem, Belgium) and Chromotrope 2R (21.3 mM; Sigma-Aldrich, Diegem, Belgium). A counterstaining was performed by the application of hematoxylin during 15 s on the sections. The number of positive cells was counted in 5 nonoverlapping high-power fields per slide, one slide per animal, and expressed as the number of chromotrope-2R-positive cells per area of lamina propria.

### 2.6. Immunohistochemistry

Mast cells were stained with an anti-mast cell protease 2 (MCPT2) antibody (1:500; Moredun Scientific, Penicuik, Scotland, UK). After deparaffinization and rehydration, endogenous peroxidase activity was inhibited with 5% H202 for 30 min and nonspecific binding sites were blocked with a protein-blocking solution (Dako, Glostrup, Denmark). Slides were incubated overnight with MCPT2 antibody at 4 °C, then 1 h at room temperature with a horse anti-mouse biotinylated secondary antibody (1:200; Vector Laboratories, Burlingame, CA, USA). Finally, slides were incubated with 3,3′-diaminobenzidine (DAB substrate development kit, Vector Laboratories) for 3 min. The number of MCPT2-positive cells was counted in 5 nonoverlapping high-power fields per slide, 1 slide per animal, and expressed as the number of MCPT2-positive cells per area of lamina propria.

### 2.7. Protein Extraction and β-Hexosaminidase Assay

Total protein was extracted for jejunum and colonic mucosa and submucosa and concentration was determined with a Bicinchoninic acid assay (BCA) test (Thermofisher, Waltham, MA, USA), as previously described [15]. To determine the beta-hexosaminidase activity, as a marker of mast cell activation [20,21], 50 µL of colonic and jejunal protein extract was incubated for 90 min with 100 µL of poly-N-acetylglucosamine dissolved in phosphate buffer at 37 °C, 5% CO_2_. The reaction was stopped by adding 0.4 M Glycine and absorbance at 405 nm was measured using FLUOstar Omega Microplate Reader (BMG Labtech, Ortenberg, Germany). The activity of beta-hexosaminidase is reported as activity units/µg of protein. 

### 2.8. Eosinophil Peroxidase

The eosinophil peroxidase (EPO) activity was determined in jejunum and colonic mucosa and submucosa by a spectrophotometric assay. The assay is based on the conversion of o-Phenylenediamine (OPD) to its colored oxidized form in the presence of H_2_O_2_ [16]. Briefly, the mucosa and submucosa were separated from the seromuscular layers by gentle dissection and immediately snap-frozen in liquid nitrogen. Extraction of EPO was performed by homogenization of the tissue in a 0.5% hexadecyltrimethylammonium chloride (Sigma-Aldrich, Diegem, Belgium) buffer at pH 6.0 followed by two freeze–thaw cycles and a centrifugation step (14,000 *g*, 30 min, 4 °C). Reaction mix containing OPD (3 mM), Potassium Bromide (6 mM), and H_2_O_2_ (8.8 mM) in HEPES buffer (50 mM) at pH 6.5 was added to the supernatant. After 1 min, the reaction was stopped with H_2_SO_4_ (4 M), and absorbance was read at 492 nm. Human EPO (Sigma-Aldrich, Diegem, Belgium) was used as a standard. Data are expressed in µg/g tissue.

### 2.9. Real-Time PCR

After separation of the mucosa and submucosa from the underlying neuromuscular layers by gentle dissection, the tissue was homogenized and extracted in TRIzol reagent (Invitrogen, Ghent, Belgium). cDNA was synthesized from 2 µg RNA using the qScript cDNA Supermix kit (Quanta Biosciences, Gaithersburg, MD, USA). The real-time PCR reaction was performed on a LightCycler 480 system with SYBR Green I Master mix (Roche Diagnostics, Vilvoorde, Belgium) with primers described in Appendix A.

qPCR was performed with 500 ng of cDNA for colonic and jejunal mucosa and 1 µg for colonic neuromuscular layers. Specific primers for sensitivity-, permeability- and inflammation-related genes were designed to cover an exon–exon junction to avoid genomic DNA amplification using NCBI Primer-Blast (https://www.ncbi.nlm.nih.gov/tools/primer-blast/) (accessed on 6 January 2020). A three-step amplification program was used: 95 °C for 10 min followed by 45 cycles of amplification (95 °C for 10 s, 60 °C for 15 s, 72 °C for 10 s) and followed by a melting curve program. Target mRNA expression was quantified relative to the housekeeping gene GAPDH using the −2ΔΔCt method.

### 2.10. Statistical Analysis

For colonic sensitivity experiments, an F-test was used to compare slopes and intercept of linear regression curves, modeling the visceromotor response (VMR) in response to wall tension. Differences between two groups were tested by two-tailed unpaired Student’s *t*-tests or Mann–Whitney U tests depending on the distribution of the data, which was checked by a Kolmogorov–Smirnov test. Differences between more than two groups were evaluated by ANOVA with a post hoc *t*-test and Sidak correction for multiple testing or Kruskal–Wallis test with Dunn’s correction for multiple comparison, when appropriate. Data are presented as mean ± SEM or median (IQR) depending on the distribution. Data were analyzed using Prism 8.3.1 (GraphPad Software, San Diego, CA, USA). The level of statistical significance was set at *p* = 0.05. 

### 2.11. Ethical Approval

All animal experiments were approved by the ethics committee for animal experiments at the University of Leuven (Protocol 1: P128/2017, protocol 2 and 3: P179/2018).

## 3. Results

### 3.1. Protocol 1: Effect of Maternal Separation on Gastrointestinal Features of the Adult BBDP-N Rat

#### 3.1.1. Stress Induces Visceral Sensitivity in Females Only

In this first section, we investigated the effect of MS on colonic sensitivity in our model. No effect of maternal separation (MS) on colonic sensitivity was found for the males (*p* = 0.99; Figure 1A; *n* = 6 per group), while females displayed an increased visceromotor response (*p* = 0.01; *n* = 9–8 per group; Figure 1B), indicating an increased pain sensitivity to visceral stimulation. When compared with MS males, MS females showed an increased visceromotor response (*p* < 0.05; Figure 1D); meanwhile, no difference between sexes was observed in the nonhandled (NH) condition (*p* = 0.75; Figure 1C), indicating an MS-induced colorectal hypersensitivity in females only.

#### 3.1.2. Stress Triggers Intestinal Inflammatory Changes in Females Only

We further assessed the infiltration of eosinophils and mast cells in both the jejunal and colonic mucosa. For both immune cell types, only females displayed MS-related alterations: maternally separated females showed an increase in eosinophil infiltration in the jejunum compared with MS males (1086 (901–1378) vs. 283(252–542) eosinophils/mm^2^; *p* < 0.05; *n* = 4–6; Figure 2A), while in NH animals, significance was lost after multiple correction testing (*p* = 0.05; *n* = 8 and 5, respectively; Figure 2A). The activity of eosinophils assessed with eosinophil peroxidase (EPO) activity was increased in MS vs. NH females (17.86 (16.2–24.6) vs. 4.4 (2.4–7.5) µg EPO/g of tissue; *p* < 0.01; *n* = 7–8; Figure 2B). Eosinophil density in the colon was not affected by MS in males while it was increased in MS vs. NH females (196 (173–267) vs. 102 (91–105) eosinophils/mm^2^; *p* < 0.001; *n* = 7–6; Figure 2C) with similar EPO activity (*p* = 0.22; Figure 2D; *n* = 8–6).

Mast cells were increased in the jejunum of NH females vs. males (1789 ± 100.9 vs. 1199 ± 80.8 mast cells/mm^2^; *n* = 6–5; *p* < 0.01; *n* = 6–5 Figure 3A) without changes in activation status, quantified by the β-hexosaminidase activity (*p* = 0.1; Figure 3B). Maternally separated females also displayed an increased mast cell infiltration vs. MS males (1727 ± 128.1 vs. 785.2 ± 80.51/mm2; *p* < 0.01; *n* = 8–5; Figure 3A), associated with an increased β-hexosaminidase activity compared with males (9.7 (6.5–13.6) vs. 5.2 (4.7–5.2) activity/µg of protein; *n* = 5–3; *p* < 0.05; Figure 3B). However, no differences between females NH and MS were reported. Colonic mast cells were significantly lower in NH females vs. males (117 ± 12.72 vs. 327 ± 33.86 cells/mm^2^; *p* < 0.001; *n* = 9–6; Figure 3C), while MS females had a mast cell density comparable to MS males, and an increased density compared with NH females (343 ± 19.97 vs. 117 ± 12.72 mast cells/mm^2^; *p* < 0.001; *n* = 8–9; Figure 3C); meanwhile, there was no effect of stress on the mast cell numbers in males. Similarly, in the colon, the β-hexosaminidase activity was only increased in MS females compared with NH (19 (14.3–22.1) vs. 11.1 (8.5–13.6) β-hexosaminidase activity/µg of protein; *p* < 0.05; *n* = 7 per group; Figure 3D), and no differences were observed in males (*p* = 0.4; Figure 3C,D). Our results so far suggest that MS triggers GI alterations in female BBDP-N rats only, which led us to hypothesize that sex hormones, and more particularly estrogen, might be involved in stress-exacerbated colonic sensitivity and eosinophil and mast cell infiltration in this spontaneous model of DGBI.

### 3.2. Protocol 2: Estrogen Involvement in Maternal-Separation-Induced GI Alterations in Adult BBDP-N Rats

#### 3.2.1. β-Estradiol Is Involved in Stress-Induced Visceral Hypersensitivity in Females

To confirm our hypothesis that the sex differences observed in the previous protocol are directly linked to the sex hormones—more particularly, estrogen—MS female rats were studied after ovariectomy (ovx) or sham surgery. Two weeks after sham surgery, colonic sensitivity was comparable to baseline (*p* = 0.65; Figure 4A; *n* = 6), while ovariectomized females displayed a decrease in the VMR in response to isobaric distensions, suggesting a decrease in visceral sensitivity (*p* < 0.001; Figure 4B; *n* = 8), which was also observed when compared to sham surgery (*p* < 0.01; Figure 4C). As shown in Figure 4D, 7 out of 8 animals demonstrated a decreased visceromotor response after ovariectomy (*p* < 0.05; *n* = 8; Figure 4D).

However, both estrogens and progesterone are depleted with ovariectomy. To confirm that the effects seen in our model were due to estrogen, we re-injected ovariectomized females with either vehicle or β-estradiol. While vehicle-treated females displayed a decreased visceromotor response compared with before the injections (MS + ovx + vehicle vs. MS + ovx; *p* < 0.01; *n* = 6; Figure 5A), we observed, in contrast, that the β-estradiol-treated ovariectomized females had an increased response compared with before injections (MS + ovx + β-estradiol vs. MS + ovx; *p* = 0.001; *n* = 6; Figure 5B), which was present in 5 out of 6 females (Figure 5D), and also compared to vehicle-treated females (*p* < 0.0001; Figure 5C). 

#### 3.2.2. β-Estradiol Increases Colonic Sensitivity in Stressed Males

We also evaluated whether the MS female phenotype could be reproduced in MS males by β-estradiol injections. No differences were observed in the vehicle-treated group before vs. after treatment (*p* = 0.4; *n* = 7; Figure 6A); meanwhile, the β-estradiol-treated males showed an increased visceromotor response compared with baseline (*p* < 0.001; *n* = 7; Figure 6B–D) in 6 out of 7 rats (Figure 6D), and when compared to vehicle-treated rats (*p* < 0.001; Figure 6C), indicating an increased colonic sensitivity in stressed males receiving β-estradiol injections.

#### 3.2.3. Stress-Induced Inflammatory Changes in Females Are Not Reversed by Ovariectomy While β-Estradiol Induces Inflammatory Changes in Stressed Males

As estrogen is known to have a direct effect on immune cells, we also investigated whether estrogen administration in males or depletion in females by ovariectomy influenced the presence and activation of intestinal eosinophils and mast cells. While in the females, ovariectomy did not impact the density of eosinophils or mast cells either in the jejunum or the colon (Figure 7A–D), we observed that β-estradiol-treated MS males presented an increased density of eosinophils in the jejunum (1257 ± 1281 vs. 527 ± 107 eosinophils/mm^2^; *n* = 5–6 per group; *p* < 0.05 Figure 7A) and a numerical but not significant increase in mast cells. In the colon, we observed a significant increase in both cell types (258 ± 69.7 vs. 98.9 ± 8.7 eosinophils/mm^2^; *p* < 0.05; *n* = 5–7; Figure 7C and 377 (277–984) vs. 153 (131–291) mast cells/mm^2^; *p* < 0.05; *n* = 5–7; Figure 7D). 

However, no differences were found in the EPO or β-hexosaminidase activity for either tissue (Appendix A). 

No effect of ovariectomy was observed in females regarding the gene expression of selected inflammatory markers in the jejunum and colon mucosa (Table 1 and Table 2; *n* = 3–7 and *n* = 6–7, respectively). 

#### 3.2.4. Β-Estradiol Induces Gene Expression of Inflammatory-Related Proteins in the GI Tract

To further substantiate the inflammatory response induced by estradiol, we measured expression of a focused set of genes related to inflammation. In β-estradiol-treated males, we observed an increased expression of F2rl1 and ccl11, coding for protease-activated receptor 2 (PAR2) and eotaxin-1 (Table 1; *n* = 4–7). In the colonic mucosa, only the expression of mcpt2, coding for mast cell protease 2, had a trend for upregulation in β-estradiol-treated males (Table 2; *n* = 4–7). 

#### 3.2.5. Gene Expression of the Estrogen Receptor GPER in the GI Tract

The estrogen receptor GPER has been shown in previous studies to be involved in gastrointestinal sensitivity. We observed an increased expression of the GPER receptor in the jejunal mucosa in female MS vs. NH BBDP-N (2.4 ± 0.4 vs. 1 ± 0.3; *p* < 0.05; *n* = 7 and *n* = 6; Table 3) and a decreased expression in the colonic mucosa of MS ovariectomized vs. sham BBDP-N (0.4 ± 0.2 vs. 1.1 ± 0.2; *n* = 6 and *n* = 7; *p* = 0.05; Table 3). 

Moreover, the expression of GPER positively correlated with the AUC of the colonic sensitivity for the MS females ovx and sham (r = 0.58; *p* = 0.01; Table 3), while no changes were observed in males.

## 4. Discussion

The previous studies of our lab characterized the BBDP-N rat as an insightful model of DGBI, but were limited to males; therefore, they lacked translatability to human DGBI, as women are known to be twice as affected by DGBI than men and more sensitive to stress and anxiety [22,23]. 

In the current study, we included females as well as males for the first time, investigating the impact of an early-life stressor in both sexes in our spontaneous rat model, with a particular focus on peripheral gastrointestinal alterations, including eosinophil and mast cell infiltration and activation, colonic sensitivity, and the role of β-estradiol in these stress-related intestinal alterations. As the effect of early-life stress on DGBI-prone animal models has never been investigated, our study brings novel insights about the interaction between sex hormones and stress in this context. Our results demonstrated that—in this animal model, with spontaneous intestinal alterations comparable to those found in patients with DGBI—an early-life stressor led to changes in gastrointestinal physiology in females only, characterized by an increased immune cell infiltration and activation in the jejunum vs. males in the same conditions, and in the colon when compared to nonhandled females, as well as an increased colonic sensitivity. Then, we demonstrated that a female GI phenotype could be induced in estrogen-treated males and part of the DGBI-like phenotype in females could be rescued by ovariectomy, indicating a critical role of estrogens in the alterations observed in the two previous studies. This was further confirmed by a reappearance of the female phenotype after the reinjection of β-estradiol in ovariectomized females.

Epidemiological observations among DGBI patients consistently reveal a sex difference that strongly suggests an interaction of the gonadal hormones in pain-processing pathways [24], as well as in the recruitment and activation of immune cells, such as mast cells. The latter, once activated, release mediators including nerve growth factor (NGF), well known for triggering neuronal excitation and pain transmission [25]. This suggests a peripheral pro-nociceptive, mast-cell-dependent role for estrogens, corroborated by other studies showing a release of histamine and serotonin by mast cells, in both animal models and DGBI patients [1,26,27,28]. 

In accordance with human studies, our results demonstrated that no baseline sex differences could be found in pain perception between males and females, although the females were in the proestrus/estrus phase. In stressed rats, only females displayed an augmentation of colonic sensitivity, which was rescued by ovariectomy [5] and reintroduced by administration of β-estradiol [8], indicating that stress-related hypersensitivity in females is related to β-estradiol in our context. The fact that β-estradiol injections triggered the development of visceral hypersensitivity reinforces its role in pain transmission. 

Several studies have linked acute-stress-induced hypersensitivity to estrogen inducing mast cell degranulation in the colon through the GPER receptor, and have linked colonic sensitivity with the expression of this receptor in IBS patients [4,12]. However, other studies have not found a different expression of GPER in female IBS vs. controls [29] or even an opposite result [30]. We observed no changes for the expression of this receptor in females MS vs. NH but a decrease in ovariectomized females compared with sham, which was positively correlated with the AUC of visceromotor response in these two groups, suggesting a link between the expression of this receptor in reduced colonic sensitivity after ovariectomy. However, in males injected with β-estradiol, no changes were observed, suggesting more a differential activation rather than expression. Discrepancy between our results and previously published results about the role of estrogen and *gper* receptor expression might also be explained by our model. We chose an early-life stress model, which probably induces long-term developmental alterations or epigenetic changes in both neuronal and GI tissues, which are still poorly characterized [31]. Moreover, maternal separation has a different impact depending on the strain used [32]. A further exploration of maternal-separation-induced changes in immune cells profiles, and the activation pattern according to sex, in addition to its effect on the development of spontaneous GI alterations, would provide further insight in this complex model, but this was beyond the scope of the current study. 

Besides their actions on the pain perception pathways, estrogens also act on gastrointestinal epithelial and immune cells, on which their receptors are expressed [9]. Mast cells have been found to be increased in female DGBI patients [17,33] and modulate the immune response. In our study, MS triggered an increased mast cell infiltration and activation in the female colon, as well as eosinophil activation in the jejunum, cooccurring with colonic hypersensitivity. This phenotype was not rescued by the ovariectomy. In fact, we observed a comparable density of immune cells in both ovariectomized and sham-operated females. It is important to note that, in our female cohort, ovariectomy was performed after puberty, while in some other studies, ovariectomy is performed at an earlier stage, such as in the Zielinska et al. study [30]. By performing the ovariectomy before the acute stress, they could prevent stress-induced mast cell infiltration in the mucosa. Our results suggest that the ovariectomy did not prevent stress-induced infiltration and activation/degranulation of immune cells when performed after the early-life stress, and after the onset of puberty. However, injections of β-estradiol in males induced an increased infiltration of eosinophils and mast cells without triggering their activation/degranulation. Taken together, those results may suggest that β-estradiol might be involved in the onset of immune cell recruitment, but not involved in the maintenance of the inflammatory response. Nevertheless, β-estradiol-treated males showed an increased expression of inflammatory markers in the jejunum and *mcpt2* in the colon, which can be released without degranulation [34], suggesting that estrogen might also induce the release of inflammatory mediators in the absence of degranulation, or a longer exposure would be needed. In the group of ovariectomized females, we observed a decrease in the visceral hypersensitivity, without any effect of the ovariectomy on immune cells density and/or activation. These results would argue for a role of estrogen at the neuronal level, either spinal or supraspinal, of the pain transmission, as previously suggested [25]. 

Nevertheless, our study also comes with some limitations. This study focused mainly on morphological evidence, providing a solid basis for further functional studies. As estrogens receptors are present in both the digestive tract and the central and peripheral nervous system, further experiments are needed to unravel whether estrogens act more through their effect on the brain or through their effect on the GI tract. As previously demonstrated in humans, the pattern of brain activation is different in men and women, which can serve as evidence for an impact of estrogen at the level of the central nervous system. 

As recently demonstrated, microbiota also play an important role in the transmission of pain and immune cells recruitment and activation, as well as playing a role in stress [35,36]. A detailed analysis of the intestinal and fecal microbiota in a BBDP-N animal model would bring important insights on the mechanism involved in maternal-separation-induced gut–brain axis alterations in our spontaneous model for DGBI. 

## 5. Conclusions

In conclusion, sex should be taken into consideration both for healthy individuals having DGBI-like symptoms under stress and for DGBI patients. Our study demonstrated a crucial role for estrogen in the triggering of visceral hypersensitivity and alterations in intestinal function of an animal model predisposed to GI (Figure 8) alterations, submitted to an early-life stress, similar to what has been observed in DGBI patients.

## Figures and Tables

**Figure 1 cells-11-01214-f001:**
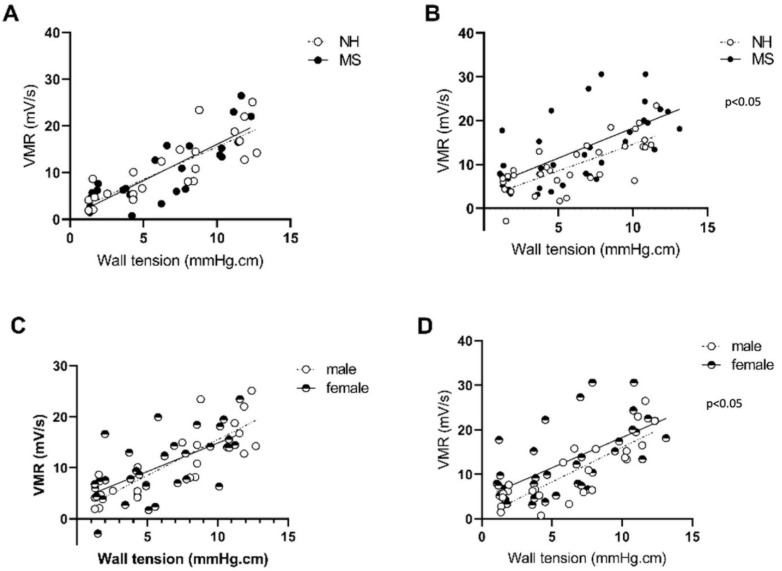
Colonic sensitivity in male and female controls and BBDP-N after maternal separation. Colonic sensitivity was assessed by testing the elevation of linear regression curves using a Fisher test between male NH (*n* = 6) and MS (*n* = 6) (**A**), female NH (*n* = 8) and MS (*n* = 9) (**B**), male NH and female NH (**C**) and male MS and female MS (**D**). NH—nonhandled; MS—maternal separation; VMR—visceromotor response. *n* = 6–10 per group.

**Figure 2 cells-11-01214-f002:**
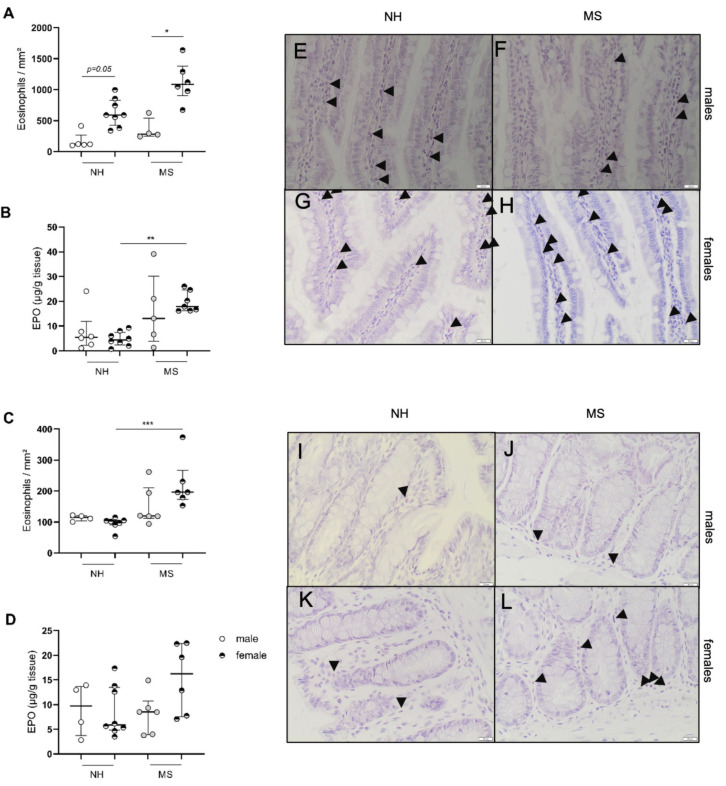
Eosinophil infiltration and activation in the jejunum and colon after maternal separation. Eosinophils stained by chromotrope 2R in the jejunum (**A**) and the colon (**C**) of BBDPN males and females. Activation of eosinophils assessed by enzymatic assay quantifying the eosinophil peroxidase in the jejunum (**B**) and colonic mucosa (**D**). Representative pictures of the jejunum for male NH (**E**); female NH (**F**); male MS (**G**); female MS (**H**) and for the colon for male NH (**I**); female NH (**J**); male MS (**K**); female MS (**L**). Eosinophils are indicated by arrows. Data are presented as median ± IQR. * *p* < 0.05; ** *p* < 0.01; *** *p* < 0.001. NH—nonhandled; MS—maternal separation. *n* = 4–8 per group.

**Figure 3 cells-11-01214-f003:**
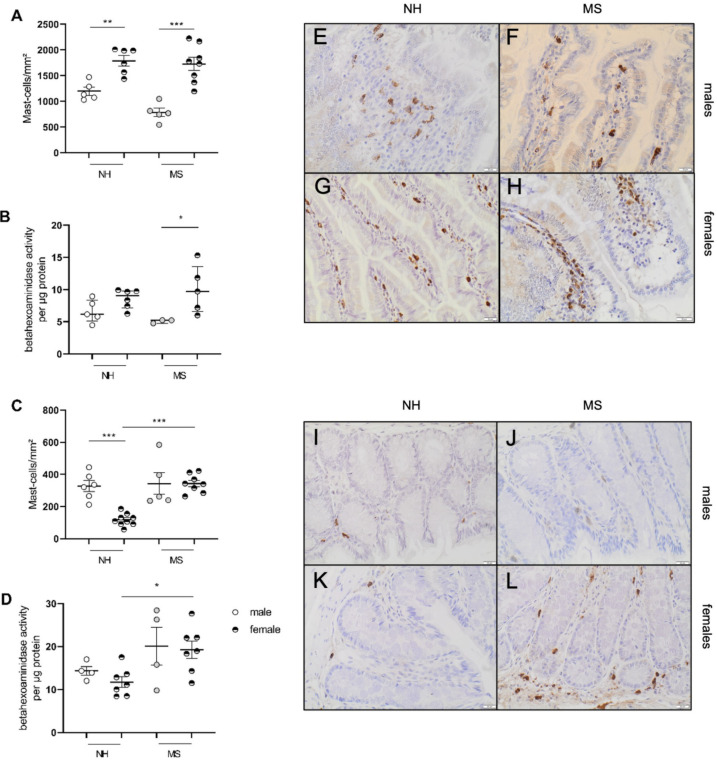
Mast cell infiltration and activation in the jejunum and colon after maternal separation. Mast cells stained by anti-protease II in the jejunum (**A**) and the colon (**C**) of BBDPN males and females. Activation of mast cells assessed in the jejunum (**B**) and colonic mucosa (**D**). Representative pictures of the jejunum of male NH (**E**); female NH (**F**); male MS (**G**); female MS (**H**) and for the colon of male NH (**I**); female NH (**J**); male MS (**K**); female MS (**L**). Data are presented as mean ± SEM (**A**,**C**) or median ± IQR (**B**,**D**). * *p* < 0.05; ** *p* < 0.01; *** *p* < 0.001. NH—nonhandled; MS—maternal separation. *n* = 3–9 per group.

**Figure 4 cells-11-01214-f004:**
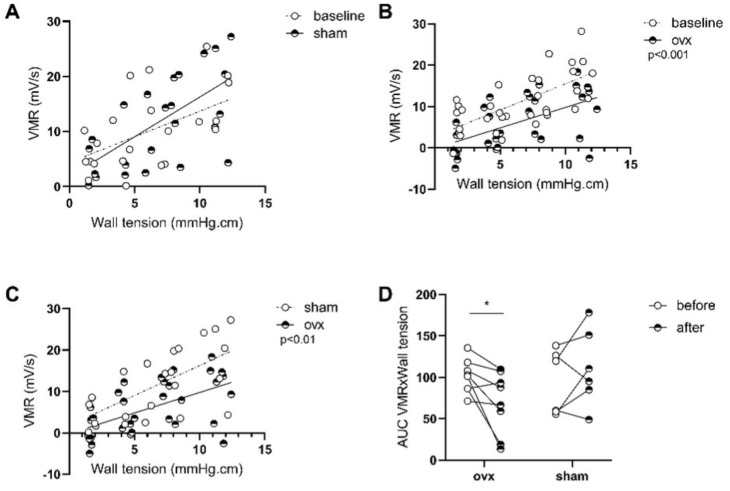
Effect of ovariectomy in MS females on colonic sensitivity. Colonic sensitivity was assessed by testing the elevation of linear regression curves using a Fisher test between MS female at baseline and 15 days after sham operation (*n* = 6) (**A**), between MS females at baseline and after ovariectomy (*n* = 8) (**B**), and between MS female after sham operation and MS female after ovariectomy (**C**). Individual representation of the effect of the ovariectomy (**D**), tested with a two-way ANOVA comparing paired time point. Data are presented as mean ± SEM (**A**). * *p* < 0.05. MS—maternal separation; ovx—ovariectomy; VMR—visceromotor response. *n* = 6–8 per group.

**Figure 5 cells-11-01214-f005:**
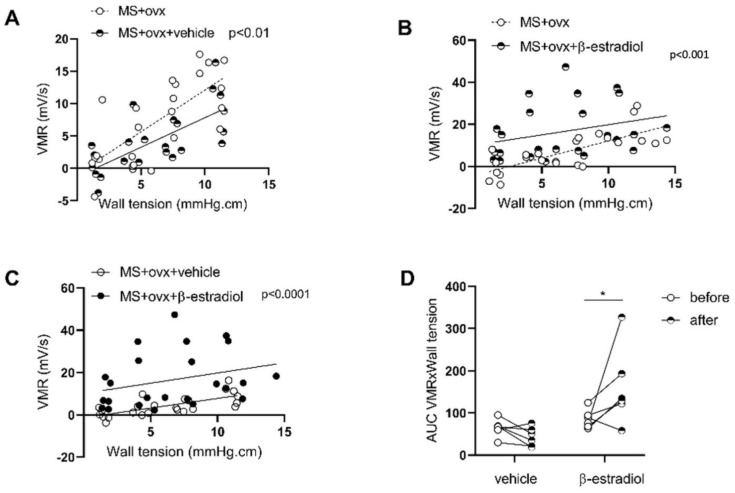
Effect of estrogen supplementation in ovariectomized MS females on colonic sensitivity. Colonic sensitivity was assessed by testing the elevation of linear regression curves using a Fisher test between MS ovariectomized females before and after injections of vehicle (*n* = 6) (**A**), between MS ovariectomized females before and after injections of β-estradiol (*n* = 6) (**B**) and between MS ovariectomized females injected with vehicle or β-estradiol (**C**). Individual representation of the effect of the treatment tested with a two-way ANOVA comparing paired time point (**D**). * *p* < 0.05. MS—maternal separation; ovx—ovariectomy; VMR—visceromotor response. *n* = 6 per group.

**Figure 6 cells-11-01214-f006:**
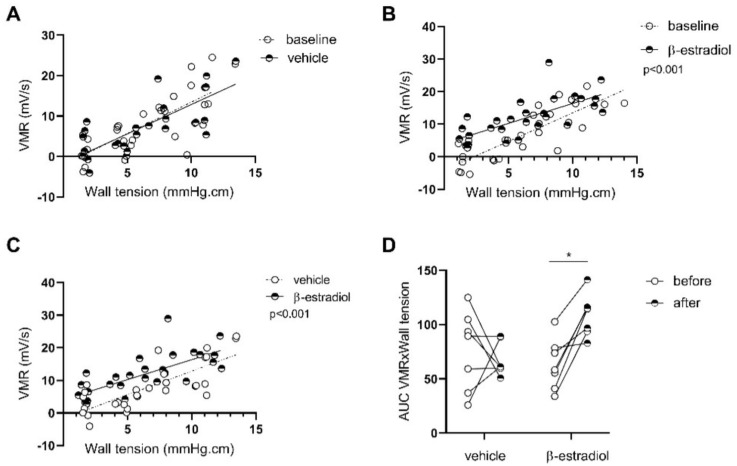
Effect of estrogen in MS males on colonic sensitivity. Colonic sensitivity was assessed by testing the elevation of linear regression curves using a Fisher test between MS males before and after 3 injections of vehicle (*n* = 7) (**A**), between MS males before and after injections of β-estradiol (*n* = 7) (**B**), and between MS males injected with vehicle and MS males injected with β-estradiol (**C**). Individual representation of the effect of the treatment tested with a two-way ANOVA comparing paired time point (**D**). * *p* < 0.05. VMR—visceromotor response. *n* = 7–10 per group.

**Figure 7 cells-11-01214-f007:**
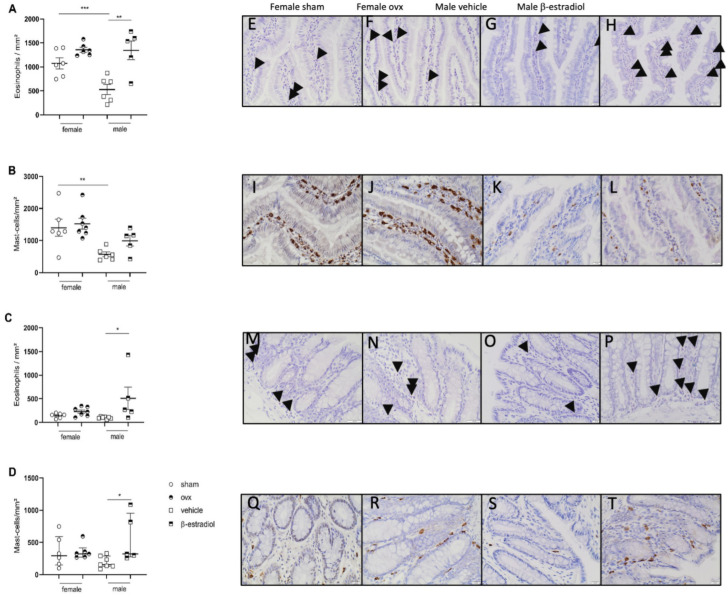
Effect of estrogen on immune cell infiltration in the jejunum and colon after maternal separation. Eosinophils stained by chromotrope2R in the jejunum (**A**) and the colon (**C**) of MS BBDP-N males injected with vehicle or β-estradiol and in females after sham surgery or ovariectomy. Mast cells stained by immunostaining in the jejunum (**B**) and colonic mucosa (**D**). Representative pictures of eosinophils infiltration in the jejunum for female sham (**E**); female ovx (**F**); male vehicle (**G**); male with β-estradiol (**H**) and for the colon for male NH (**M**); female NH (**N**); male MS (**O**); female MS (**P**). Representative pictures of mast cells infiltration in the jejunum for female sham (**I**); female ovx (**J**); male vehicle (**K**); male with β-estradiol (**L**) and for the colon for male NH (**Q**); female NH (**R**); male MS (**S**); female MS (**T**). Eosinophils are indicated by arrows. Data are expressed as mean ± SEM or median with IQR (**D**) * *p* < 0.05; ** *p* < 0.01; *** *p* < 0.001. ovx—ovariectomy. *n* = 5–7 per group.

**Figure 8 cells-11-01214-f008:**
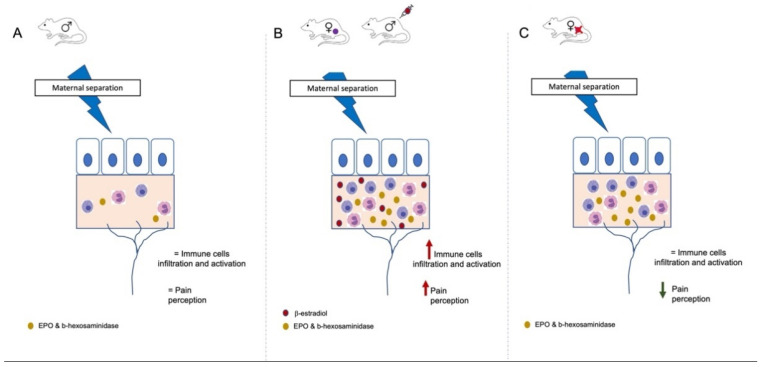
Summary of results for both sexes under stress conditions. In the BBDP-N rat, an animal prone to develop DGBI, early-life stress did not change either the immune cell infiltration and activation in the GI tract or colonic hypersensitivity in males (**A**). In females and males injected with β-estradiol, an increase in immune cells infiltration and activation in the GI tract was associated with an increased colonic sensitivity (**B**). In ovariectomized females, the immune cell infiltration and activation did not change, while the perception of colonic distention was reduced (**C**).

**Table 1 cells-11-01214-t001:** Effect of β-estradiol on gene expression of jejunal mucosal inflammatory markers.

Gene	Males Vehicle	Males β-Estradiol	*p*-Value	Females Sham	Females Ovx	*p*-Value
F2rl1	1.3 ± 0.4 *n* = 6	2.4 ± 0.3 *n* = 6	0.05	0.7 (0.4–4.1) *n* = 5	0.9 (0.2–5.3) *n* = 7	0.87
mcpt2	1.3 ± 0.5 *n* = 5	2.4 ± 0.5 *n* = 7	0.17	2 ± 0.99 *n* = 6	10.8 ± 4 *n* = 7	0.24
ccl2	2.3 (0.1–6.1) *n* = 5	1 (0.1–1.8) *n* = 6	0.32	1.4 (0.1–9.9) *n* = 4	2 (0.9–4.5) *n* = 6	0.76
ccl11	2.3 (0.1–4.3) *n* = 6	5.1 (4–6.1) *n* = 7	0.02	1.4 ± 0.6 *n* = 5	3.5 ± 1.2 *n* = 8	0.15
il4	0.43 (0.05–2.4) *n* = 4	6.8 (1.3–8.8) *n* = 3	0.1	0.2 (0.1–0.3) *n* = 5	0.06 (0–4.6) *n* = 6	0.83

ovx—ovariectomy; F2rl1—gene coding for par2 (protease-activated receptor 2); mcpt2—mast cell protease 2; ccl2—gene coding for mcp1 (monocyte chemoattractant protein1); ccl11—gene coding for eotaxin-1; il-4—interleukin 4. Data are expressed as mean ± SEM or median (IQR) depending on the distribution. Unpaired *t*-test or Mann–Whitney depending on the distribution.

**Table 2 cells-11-01214-t002:** Effect of β-estradiol on gene expression of colonic mucosal inflammatory markers.

Gene	Males Vehicle *n* = 4	Males β-Estradiol *n* = 7	*p*-Value	Females Sham *n* = 6–7	Females Ovx *n* = 7	*p*-Value
F2rl1	1 (0.7–1.3)	1.4 (0.5–2)	0.31	1.1 ± 0.2	0.8 ± 0.2	0.28
mcpt2	1 (0.8–1.1)	4.1 (3.1–18.9)	0.07	1.7 ± 0.6	0.7 ± 0.2	0.15
ccl2	0.8 (0.7–1.6)	3.6 (0.3–10.4)	0.31	1.6 ± 0.6	4.8 ± 2.6	0.28
ccl11	1.1 (0.6–1.4)	1 (0.7–2.4)	0.99	1.7 ± 0.7	0.6 ± 0.2	0.18

ovx—ovariectomy; F2rl1—gene coding for par2 (protease-activated receptor 2); mcpt2—mast cell protease 2; ccl2—gene coding for mcp1 (monocyte chemoattractant protein1); ccl11—gene coding for eotaxin-1. Data are expressed as mean ± SEM or median (IQR) depending on the distribution. Mann–Whitney *t*-test for males and Unpaired *t*-test for the females.

**Table 3 cells-11-01214-t003:** Gene expression of the estrogen receptor *gper1* in jejunum and colon mucosa.

	Females				Males	
Group	NH	MS	MS Sham	MS Ovx	MS Vehicle	MS β-Estradiol
jejunum	1 ± 0.3 *n* = 6	2.4 ± 0.4 *n* = 7	1.3 ± 0.4 *n* = 7	1.6 ± 0.6 *n* = 7	0.9(0.4–3.7) *n* = 6	0.7(0.3–2.2) *n* = 7
*p*-value	0.03		0.66		0.73	
colon	0.7(0.3–0.7) *n* = 5	0.5(0.1–2.9) *n* = 7	1.1 ± 0.2 *n* = 7	0.4 ± 0.2 *n* = 6	0.9 ± 0.3 *n* = 5	2.2 ± 0.6 *n* = 6
*p*-value	0.99		0.05		0.24	
Correlation with VMR AUC	r = −0.06 *p* = 0.82		r = 0.58 *p* = 0.01		r = 0.26 *p* = 0.42	

AUC—area under the curve; VMR—visceromotor response; NH—nonhandled; MS—maternal separation; BBDPN—BioBreeding diabetes-prone normoglycemic; ovx—ovariectomy. Unpaired *t*-test or Mann–Whitney depending on the distribution.

## Data Availability

The data presented in this study are available on request from the corresponding author.

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
