# Peer review of "Estrogens Play a Critical Role in Stress-Related Gastrointestinal Dysfunction in a Spontaneous Model of Disorders of Gut–Brain Interaction"

_cells, 2022, doi:10.3390/cells11071214_

Round 1
Reviewer 1 Report
Accarie et al present results on an extremely interesting and timely topic: relationship between sexual hormones and gastrointestinal symptoms in a stress-induced model. Experimental design is accurate and beta-estradiol injection experiments are well-oriented. However, some revisions are needed in order to make a stronger manuscript to be published in a Q1 journal.
- Introduction: what do authors mean for disorders of gut-brain interaction (DGBI). Please, could you indicate some of them, with specific examples?
- Figures and Figure Legends (also for Supplementary Fig) are difficult to understand and to read. Authors should organize them by alphabetical order (A, B, C, D), from left to right.
- Results are difficult to follow. Please, include more subsections that describe in a declarative manner the objective and results for each experiment. For example, instead of organizing just by the two main protocols, I suggest to subdivide this section based on main results.
- Many graphs are given, but no photographs or images are presented. This is an important point to be resolved, since the manuscript relays its strength, mainly, on morphological data.
- Discussion section should start with a clear summary of the objetives and results. +
- A “limitations of the study” section would be highly appropriate, since this work does only present morphological and PCR gene expression data, and no functional, microbiota-related, or neuronal-related experiments and results are performed (which, definitively, should be mention as next steps and perspective). Authors do not measure any brain parameters, but they talk about DGBI. Do estrogens act directly on the gastrointestinal tract or might they be acting through theirs effects on brain? Due to the implications between maternal separation and microbiota-gut-brain axis alteration, how do authors explain that they do not evaluate this main player?

Author Response
We thank the reviewer for their constructive comments and have revised the manuscript accordingly. Please find our point-by-point reply in attachment.

Reviewer 2 Report
This is a nicely performed and described set of studies that substantially extends our understanding of the interplay between sex, estrogen, and early life stress in this model. I have only minor suggestions.
- The discussion is a bit long and could perhaps be trimmed by 10-20%.
- It might be nice to add a schematic summarizing the results and the scheme of interaction between these factors and the end points that the authors have studied.
Author Response
We thank the reviewer for their constructive comments and have adapted the manuscript accordingly. Please find our point-by-point responses in attachment.
